# Could Interleukin-33 (IL-33) Govern the Outcome of an Equine Influenza Virus Infection? Learning from Other Species

**DOI:** 10.3390/v13122519

**Published:** 2021-12-15

**Authors:** Christoforos Rozario, Luis Martínez-Sobrido, Henry J. McSorley, Caroline Chauché

**Affiliations:** 1Centre for Inflammation Research, The Queen’s Medical Research Institute, The University of Edinburgh, Edinburgh BioQuarter, Edinburgh EH16 4TJ, UK; crozario@exseed.ed.ac.uk; 2Texas Biomedical Research Institute, San Antonio, TX 78227, USA; lmartinez@txbiomed.org; 3Division of Cell Signalling and Immunology, School of Life Sciences, University of Dundee, Wellcome Trust Building, Dow Street, Dundee DD1 5EH, UK; hmcsorley001@dundee.ac.uk

**Keywords:** equine influenza virus, NS1, PB1-F2, PA-X, airway epithelial cell, cell death, interleukin-33, asthma, secondary bacterial infection

## Abstract

Influenza A viruses (IAVs) are important respiratory pathogens of horses and humans. Infected individuals develop typical respiratory disorders associated with the death of airway epithelial cells (AECs) in infected areas. Virulence and risk of secondary bacterial infections vary among IAV strains. The IAV non-structural proteins, NS1, PB1-F2, and PA-X are important virulence factors controlling AEC death and host immune responses to viral and bacterial infection. Polymorphism in these proteins impacts their function. Evidence from human and mouse studies indicates that upon IAV infection, the manner of AEC death impacts disease severity. Indeed, while apoptosis is considered anti-inflammatory, necrosis is thought to cause pulmonary damage with the release of damage-associated molecular patterns (DAMPs), such as interleukin-33 (IL-33). IL-33 is a potent inflammatory mediator released by necrotic cells, playing a crucial role in anti-viral and anti-bacterial immunity. Here, we discuss studies in human and murine models which investigate how viral determinants and host immune responses control AEC death and subsequent lung IL-33 release, impacting IAV disease severity. Confirming such data in horses and improving our understanding of early immunologic responses initiated by AEC death during IAV infection will better inform the development of novel therapeutic or vaccine strategies designed to protect life-long lung health in horses and humans, following a One Health approach.

## 1. Introduction

Equine influenza (EI) is one of the most important respiratory diseases of horses worldwide [1,2]. Outbreaks of EI disease occur recurrently despite the availability of vaccines and result in severe economic loss for the equine industry [1,3]. EI is caused by an influenza A virus (IAV), a member of the Orthomyxoviridae family, whose genome is composed of eight single-stranded RNA segments of negative polarity. Two subtypes of equine influenza virus (EIV) have been described in the horse, H7N7—thought to be extinct, and H3N8, which has been circulating in the horse population since 1963 [4,5,6,7].

Horses infected with EIV develop typical respiratory disorders, commonly seen in IAV-infected humans, including fever, nasal discharge, cough, and lethargy [1,2]. The histological changes caused by EIV are usually confined to the respiratory tissue and adjacent lymph nodes. In the respiratory tract, rhinitis and moderate to severe tracheitis and bronchitis are usually observed. They are accompanied by a loss of ciliated respiratory epithelium in infected areas, as well as a reduction in goblet cell numbers, diffuse epithelial vacuolar degeneration or necrosis, epithelial hyperplasia, or squamous metaplasia, and lymphocytic infiltration of the lamina propria from the nasal mucosa to the bronchi [8,9].

Like other IAVs, H3N8 EIV undergoes genetic evolution, and since its emergence in 1963, EIV has evolved in several lineages and sublineages, accumulating mutations in its genome [10,11,12,13]. Since the early 2000s, the dominant circulating sub-lineages around the world belong to the Florida clades 1 (FC1) and 2 (FC2). Until recently, FC2 EIVs have predominately been isolated in Europe and Asia, whilst the majority of FC1 EIVs have been isolated in North America [14]. However, in 2018 FC1 EIV strains were suddenly detected in Europe and spread widely, causing many outbreaks in Europe, USA, and Africa from late 2018 to 2019 [15,16,17]. Interestingly, evolutionary distinct EIVs display different levels of virulence, with some strains causing longer periods of pyrexia and duration of coughing [18,19], although the underlying reasons are still unclear. High viral loads can play a role in disease progression [20,21], but do not always correlate with disease severity [22,23,24,25,26], which is generally associated with excessive tissue inflammation and damage to the respiratory epithelium [27].

Data obtained in humans and mice indicate that the type of cell death that airway epithelial cells (AECs) undergo during an IAV infection impact disease severity [28,29,30]. As recently reviewed by Atkin-Smith and colleagues, during an IAV infection AECs may die by necrosis (unprogrammed or programmed necrosis—necrosis or necroptosis and pyroptosis, respectively), or apoptosis (programmed cell death via intrinsic or extrinsic pathways) [28]. Apoptosis is generally considered anti-inflammatory and is thought to prevent the release of damage-associated molecular patterns (DAMPs) while effectively eliminating virus replication [5,6,31]. Instead, necrosis is thought to cause pulmonary inflammation by allowing the release of DAMPs, such as interleukin-33 (IL-33) [30,32] (Figure 1).

Levels of inflammatory response and tissue damage during an EIV infection vary between viral strains [9,33]. Evidence obtained in mice and humans indicates that the type of cell death that AECs undergo during an IAV infection will determine the magnitude of bioactive IL-33 released and the resulting lung inflammatory response [27,34,35,36]. Here, we integrate the current knowledge obtained in horses, humans, and mouse models to discuss how viral determinants and host immune responses that control AEC cell death during an IAV infection may shape IL-33 responses and promote subsequent pulmonary complications, such as secondary bacterial infections or asthma exacerbations. The health of animals, humans, plants, and the environment is interconnected. Understanding the early immunologic response initiated by AEC death during IAV infection, using an integrated approach to health, will better inform the development of novel therapeutic or vaccine strategies designed to protect life-long lung health in humans and horses, following a One Health approach.

## 2. Determinants of Lung Damage and Pulmonary Complications during an EIV Infection

### 2.1. Viral Determinants of Airway Epithelial Cell Death

The EIV genome encodes at least 12 known proteins, including nine structural and three non-structural proteins [37,38] (Figure 2).

The structural proteins comprise constituent of the viral ribonucleoprotein (vRNP) complex, including the three subunits of the viral polymerase (PB2, PB1, and PA) encoded by segments 1, 2, and 3, respectively; and the viral nucleoprotein (NP) encoded by segment 5. In addition, the membrane-associated glycoproteins Hemagglutinin (HA) and Neuraminidase (NA) are encoded from segments 4 and 6, respectively; and the matrix protein 1 (M1) and the ion channel matrix protein 2 (M2) are encoded on segment 7 (colinear mRNA for the former and via alternative splicing for the latter). At last, the Nuclear Export Protein (NEP, formerly known as NS2) is encoded from segment 8 using an alternative splicing mechanism [39].

More importantly, the 3 non-structural proteins of EIV include the non-structural protein 1 (NS1), PB1-F2 and PA-X [11,40,41] (Figure 1). These proteins are non-essential for viral replication; however, they play an important role in viral pathogenicity, inflammation, and tissue damage during infection [11,40,41,42,43,44,45,46]. They are best known for counteracting type I interferons (IFNs), important cytokines in the host defense against viruses through the induction of antiviral effector molecules encoded by IFN-stimulated genes (ISGs) [47]. Furthermore, NS1, PB1-F2, and PA-X are also known for modulating AEC death during viral infection [11,40,41] (Figure 1). Interestingly, at the viral population level natural variants occur in these non-structural proteins that affect their regulatory functions [11,39,40,41,42,43,44,48,49].

#### 2.1.1. NS1

NS1 is encoded on a colinear mRNA of segment 8 [39] (Figure 2) and is most often a 230 amino acid-long protein [43]. However, mutations that either suppress the stop codon at position 231 or create a premature stop codon result in length variations [50]. NS1 represents a major virulence factor common to all IAVs, whose best described function is to antagonize the type I IFN pathway [51,52,53]. Polymorphisms among NS1 proteins exist [52,54,55,56], and the protein C-terminus accounts for important functional differences between viral strains and subtypes. Indeed, some NS1 proteins harbor an “ESEV/EPEV” PDZ binding motif from residues 227 to 230. This motif is usually found in IAVs that infect birds [57]. PDZ domains are protein-protein recognition modules within a multitude of proteins that organize diverse cell-signaling assemblies [58], and notably cellular apoptosis. NS1 proteins carrying the “ESEV/EPEV” motif have notably been shown to inhibit cellular apoptosis through interaction with Scribble [59,60]. In the case of EIV, virus isolates in circulation from viral emergence in 1963 until the late 1990s expressed a 230 amino acid-long NS1 protein carrying an ‘ESEV/EPEV’ motif at residue 227–230. This motif was lost after the late 1990s when a premature stop codon at position 220 was introduced in the NS1 coding sequence, which resulted in a C-terminal truncation of 11 amino acids [10,61]. Data obtained by us showed that an EIV expressing an artificially extended version of NS1 (230 amino acids) induced premature apoptosis in infected equine cells compared to one carrying the natural C-terminal truncated form of NS1 [40]. These data contrast with those obtained with avian IAVs [59,60] and may result from functional differences between IAV subtypes infecting different hosts or indicate that other viral factors aside from the ESEV/EPEV motif of full-length NS1 are necessary for PDZ binding and apoptosis inhibition. In addition, it is possible that despite carrying the ESEV/EPEV motif, the 230 amino acid-long NS1 protein of EIV cannot actually interact with Scribble to prevent apoptosis.

In IAVs found in humans (H3N2 and H1N1 IAVs), NS1s have also been shown to limit early apoptosis, by either inhibiting the type I IFN pathway which indirectly leads to inhibition of caspase-3 activation [31] or by activating the Phosphoinositide 3-kinase (PI3K)/Akt-pathway via NS1 residues Y89 and P164, subsequently inhibiting caspase 9 and limiting activation of virus-induced cell death programs [62,63,64,65,66]. In the case of EIV, residue 186 has been described as important for EIV control of type I IFNs, and polymorphism at this residue impacted EIV control of the type I IFN response [40,67], however, it did not seem to impact caspase 3 activation [67]. As for PI3K/Akt-pathway activation, EIV NS1 residues Y89 and P164 are well conserved in the EIV population [68], which suggests that EIV NS1 may be able to activate the PI3K pathway and modulate apoptosis as described for H3N2 and H1N1 IAV NS1 proteins, although this awaits experimental confirmation.

#### 2.1.2. PA-X

PA-X is an N-terminally truncated product of PA segment 3. Two natural variants of PA-X exist: the more common form of 252 amino acids, and a C-terminally truncated variant of 232 amino acids usually found in canine influenza viruses (CIV), the human H1N1 pandemic virus from 2009, and some swine influenza viruses [69]. PA-X is best known for its ability to suppress host protein synthesis and induce host mRNA degradation in infected cells, including those of the type I IFN pathway [70], and for its anti-apoptotic activity [26,41,49,71]. Of note, data obtained in mice indicate that PA-X might be accumulated in the late phase of viral infection, due to the inefficiency of frame-shifting during translation, and might exert more influence during that period [71]. In EIV, PA-X C-terminus and notably residue 231, has been shown to impact the control of type I IFNs and PA-X anti-apoptotic activity [41]. Furthermore, as seen for other IAVs, PA-X polymorphisms exist among EIV strains, which affects the protein function in infected cells. The majority of EIV strains express a protein of 252 amino acids and some strains carry a C-terminal truncated protein (232 amino acid-long). Of note, some EIV strains isolated in the United Kingdom (UK) in 2007 expressed an even shorter PA-X protein truncated of 42 amino acids at its C-terminus, e.g., A/equine/Richmond/1/2007 [13]. This truncation did not persist in the EIV population and was only found in a few other isolates in the UK in 2007 [13]. The impact of such dramatic C-terminal truncation on the protein’s functions remains elusive.

#### 2.1.3. PB1-F2

PB1-F2 is an N-terminally truncated product of PB1 segment 2. Full-length PB1-F2 proteins are 90 amino acid-long and their C-terminal region possesses a mitochondrial targeting sequence that allows them to translocate into the mitochondria and induce cellular apoptosis [48]. In EIV, the amino acid sequence of PB1-F2 varies [29], and while most of the currently circulating EIV strains express an 81 amino acid-long protein, other EIVs express a protein of 90 amino acids (e.g., A/equine/Saone-et-Loire/1/2015) [11]. Whether the latter can translocate into the mitochondria and induce apoptosis in equine cells has not been formally tested.

In mice, several other PB1-F2 motifs of H1N1 IAV have been shown to trigger cytotoxic cell death in epithelial and immune cells, such as I68, L69, and V70 [44]. In influenza A/Puerto Rico/8/1934 H1N1, PB1-F2 with amino acids I68, L69, and V70 increased mortality, promoted pneumococcal pneumonia, as well as pulmonary inflammation with the accumulation of macrophages, neutrophils, and cytokines in the bronchoalveolar lavage fluid of mice at 7 days after infection [45]. These effects were reverted with mutations I68T, L69Q, and V70G. It is not known if these motifs are important in the context of an EIV infection.

In addition, PB1-F2 has also been implicated in the imbalance of host inflammatory responses during infection in mouse models. Residue S66 has notably been shown to decrease type I IFN levels, increase pro-inflammatory cytokines, and enhance lung infiltration of immune cells [72,73]. Furthermore, in human H3N2 viruses PB1-F2 motifs L62, R75, R79, and L82 have been associated with increased morbidity and enhanced risk of secondary bacterial pneumonia, while P62, H75, Q79, and S82 have been shown to have opposite effects [44]. In the case of EIV, FC1 strains causing a severe disease have been shown to carry PB1-F2 motifs L62, R75, and R79 (e.g., A/equine/Belfond/6-2/2009). In contrast, FC2 strains that carried H75 and Q79 in PB1-F2 were associated with milder respiratory disease (e.g., A/equine/Cambremer/1/2012 and A/equine/Saone-et-Loire/1/2015) [11]. Whether EIV FC1 strains generally carry pro-inflammatory motifs in PB1-F2, while EIV FC2 strains carry non-inflammatory ones remains to be formally proven. Furthermore, although responsible for a milder respiratory disease, A/equine/Saone-et-Loire/1/2015 expressed a PB1-F2 protein with motifs I68, L69, and V70 [11]. Whether this inconsistency reflects virus subtype- or host-dependent differences remains to be proven.

Aside from viral factors, immune cells and cytokines can also promote lung inflammation and AEC death during an IAV infection, being in humans or horses.

### 2.2. Immune-Mediated Damage to the Respiratory Epithelium during EIV Infection

The extent to which anti-IAV innate and adaptive immune responses are protective before becoming pathogenic is still highly debated [74,75,76,77,78,79,80], and rarely studied in the context of an EIV infection. A more integrated approach to health to answer this question would benefit humans and horses, following a One Health approach.

Cytokines affect all aspects of the immune response against IAV, from promotion to inhibition of inflammation, priming of immune responses, and preventing excessive tissue damage [74]. Excessive cytokine signaling frequently exacerbates lung epithelial damage during IAV infection, and notably, type I and II IFNs are critical inflammatory mediators implicated in the pathogenesis of IAV [32,81,82,83]. In the case of EIV, some strains are associated with exuberant inflammatory responses, described as a “cytokine storm”, while others are not [84]. This so-called ‘storm’ is associated with excessive levels of anti-viral and pro-inflammatory cytokines, and widespread damage of the respiratory tissue, despite the control of viral replication [84,85]. This was the case for an H3N8 FC1 EIV responsible for a widespread epidemic in South African horses in 2003 [86]. Similarly, data obtained in experimental infections indicated that A/equine/Sussex/1989 was more pathogenic and associated with higher levels of type I IFNs and IL-6 in nasal secretions than A/equine/Newmarket/2/1993 [33]. How polymorphism in NS1, PB1-F2, and PA-X impacts these phenotypes remains elusive. However, it is now clear that depending on the viral strain and the host pathophysiological status, type I IFNs can contribute to immunosuppression or immunopathology [47]. Indeed, in severe IAV infections, type I IFNs have been shown to induce AEC death and cause tissue damage via induction of cluster of differentiation 95 (CD95)/Fas or TNF-related apoptosis-inducing ligand (TRAIL) by inflammatory monocytes, and death receptor 5 (DR5) expression by epithelial cells [87,88]. In mice, type I IFNs have also been shown to regulate TRAIL expression on CD8+ T cells and in turn control the magnitude of the CD8+ T cell response [47]. Cytotoxic CD8+ T cells play a central role in apoptosis induction in IAV-infected cells via perforin and granzyme-mediated cytotoxicity [89,90,91] and in the clearance of infected cells [92,93,94,95]. However, they can also contribute to lung injury by inducing ‘bystander damage’ to uninfected airway epithelial cells [96].

Natural killer (NK) cells and CD4+ Th1 cells also induce AEC apoptosis in infected cells through Fas/FasL and TRAIL/TRAIL-DR5 interactions, complementing CD8+ T cells perforin and granzyme-mediated cytotoxicity [89,90].

Type I IFNs can also stimulate type II IFN (IFN-γ) production by immune cells during IAV infection [97,98]. IFN-γ is produced by both NK cells and CD4+ Th1 cells and promotes NK and B cell interactions, which are essential for antibody production and NK cell cytotoxicity, two vital host protective mechanisms against IAV. In addition, together with interleukin-2 (IL-2), IFN-γ also promotes CD8+ T cell activation and B cell differentiation [99].

Furthermore, although macrophages and neutrophils are important in the clearance of virus-infected apoptotic cells, as well as suppressing inflammation and initiating wound repair post influenza virus infection [94,100], their excessive lung infiltration results in Influenza-induced lung injury [101,102,103].

The control of viral infection and clearance of infected cells, as well as termination of inflammation and initiation of epithelial repair programs, are essential for host survival, being humans or horses. During this recovery period, individuals are at great risk of developing pulmonary complications, such as secondary bacterial infections [104,105,106].

### 2.3. Secondary Bacterial Pneumonia Post EIV Infection

Secondary bacterial pneumonia following an EIV infection can be fatal [9,32,107,108], and horses are particularly at risk from day 7 to day 14 post-EIV infection. In horses, the most common bacteria causing complications post-EIV infection are *Staphylococcus aureus*, *Bacteroides* sp., and *Streptococcus equi subsp. zooepidemicus* (*S. zooepidemicus*), with the latter being the most frequently detected [107,109,110,111].

Several factors have been implicated in secondary bacterial infection post-EIV infection, including viral denudation of the airway epithelium and surface receptor changes that may increase bacterial colonization [112,113]. Furthermore, in mice models, altered neutrophil functions and excessive production of immunosuppressive IL-10 have also been implicated [114,115,116,117]. Additionally, alveolar macrophages (AMs) have the ability to control bacterial infections by coordinating the innate immune response via the production of pro-inflammatory cytokines, and by recruiting and scavenging apoptotic polymorphonuclear cells [118,119]. Recently, macrophage functional depression mediated by IFN-γ [120] or reduced TLR signaling in macrophages in response to bacterial ligand [115] was also found to predispose to secondary bacterial infection post-IAV infection. NK cells, whose main function is to kill IAV-infected cells, may also interact with macrophages to regulate macrophage-mediated bacterial clearance [121,122]. Accordingly, the influenza virus has been shown to impair NK cell responses and to predispose to *S. aureus* secondary infection post-influenza virus infection [123].

Many of the cytokines and chemokines responsible for inducing antibacterial effector molecules and for coordinating the protective responses against lung bacterial infections are IFN inducible (mainly through IFNγ) [124] Type I IFNs can be protective or detrimental to the host during bacterial infection in a bacterium-specific manner [47,124]. In mouse models, a protective role for type I IFNs has been reported against *Streptococcus pneumoniae* [16,125,126,127] by contributing to the optimal activation of macrophages [125]. In contrast, type I IFNs are detrimental for the host during *Staphylococcus aureus* infection [128].

As important modulators of the host type I IFN response, NS1, PB1-F2, and PA-X, may indirectly influence the development of secondary bacterial infection post-EIV infection, although little data is available. As mentioned earlier, PB1-F2 residues L62, R75, R79, and L82, notably found in EIV strains causing strong morbidity [11], have been associated with increased susceptibility to secondary bacterial pneumonia in mice [44,72,129,130], however, the underlying mechanisms remain to be fully elucidated. To our knowledge, aside from PB1-F2, how polymorphism in NS1 and PA-X impacts secondary bacterial infection post EIV infection has not been considered.

Alongside viral factors, host factors also play a role in secondary bacterial infections post-IAV infection. Indeed, evidence obtained in human and mice suggests that while the severity of IAV infection is potentiated in individuals with airway allergic inflammation, e.g., asthmatics [131,132,133,134,135,136,137], the risk of developing secondary bacterial infections seem to be lower in this population [138]. IL-33 solicits particular interest in this context, as it has been shown to be released by necrotic AECs during IAV infection in humans and mice, it regulates lung immune responses to both viral and bacterial infections, and it is also increased in the lungs of asthma patients [139,140]. In horses, little is known about IL-33, apart from the fact that horses express an IL-33 protein, which shares a certain degree of homology with the mouse and human protein [141]. Whether it is released during an EIV infection in horses, or whether asthmatic horses express higher levels of IL-33 is currently unknown. This area of research is being actively investigated by our team.

## 3. IL-33 during an IAV Infection

### 3.1. IL-33 Biology

IL-33 is an “alarmin” cytokine (Figure 3), member of the IL-1 family of cytokines, constitutively expressed by epithelial and endothelial cells of barrier sites, such as the lung, intestine, and skin [142]. The full-length IL-33 protein consists of an N-terminal nuclear domain, a protease-sensitive central domain, and a C-terminus IL-1-like receptor-binding domain (Figure 3A). IL-33 lacks the N-terminal signal peptide required for conventional secretion. Instead, IL-33 is stored in the nucleus of expressing cells, anchored to the chromatin (Figure 3B). Whether or not nuclear IL-33 affects gene expression within IL-33-expressing cells is controversial [143] and may be cell-type dependent [144,145,146]. It is well documented that upon release in the intercellular space, IL-33 acts as a potent inflammatory mediator [31,147]. In the context of viral infection, it is generally accepted that full-length IL-33 is released as a bioactive form by cells that undergo necrosis and necroptosis [148,149] (Figure 3B), as well as by cells that undergo cytolysis or release their nuclear contents [150]. The activity of IL-33 can be further increased by protease cleavage in the central domain (Figure 3C) [151]. This cleavage occurs via a range of cysteine and serine proteases, which can be intrinsic to the IL-33–expressing cell, released by recruited mast cells or neutrophils, or present in allergens [31,152]. Full-length IL-33 can also be cleaved by caspases 3 and 7 during apoptosis (Figure 3B), however, this occurs in the cytokine receptor-binding domain, which inactivates the cytokine [152]. In that sense, IL-33 differs from other members of the IL-1 cytokine family, such as IL-1β and IL-18, which are produced as inactive pro-forms and then processed into active mature forms by caspases 3 and 7, or by caspase 1, an important inflammasome cytokine [153]. In early studies, IL-33 was suggested to be activated by caspase-1 [154], while most recent studies showed no activity of caspase 1 on IL-33 [31,152,155]. Of note, in the context of pyroptosis, which is induced by the inflammasome and associated with secretion of IL-1β and IL-18 [156], evidence suggests that IL-33 may contribute to the pyroptotic-generated inflammatory response, however, whether bioactive IL-33 is released by pyroptotic cells themselves remains controversial [157,158].

IL-33 signals through a receptor complex consisting of suppression of tumorigenicity 2 (ST2) and IL-1R accessory protein (IL-1RAP—a common component of IL-1 cytokine family receptors) [147,159] (Figure 3B). A large variety of immune cells constitutively express ST2 [31,160], including CD4+ T cells, particularly Th2 cells [161] and regulatory T cells (Tregs) [162,163,164], and CD8+ T cells after IL-12 stimulation [165,166]. In addition, ST2 is also expressed on innate immune cells, such as type 2 innate lymphoid cells (ILC2s) [167,168,169,170], alternatively activated macrophages (AAMs) [171], neutrophils [172], mast cells [173,174,175], eosinophils [176], basophils [172], NK and invariant natural killer T (iNKT) cells [177]. Furthermore, stromal lung cells including endothelial cells, epithelial cells, and fibroblasts may also constitute important cellular targets of IL-33 [178,179,180,181,182], although this area of research remains largely underexplored.

Bioactive IL-33 is negatively regulated by several mechanisms (Figure 3): as mentioned earlier, the protein is stored preformed in the nucleus of expressing cells, tethered to the chromatin, which prevents its release in the intercellular space at homeostasis (Figure 3B); and IL-33 is inactivated by caspases (3 and 7) during apoptosis (Figure 3B). Additionally, the soluble form of ST2 (sST2) acts as a decoy receptor for the cytokine and prevents interaction with its receptor (ST2-IL1RAP complex) on target cells [31] (Figure 3D); finally, the cytokine is sensitive to the oxidative extracellular environment, and shortly after release (within 1 h of release) disulphide bonds are formed between cysteines residues in the receptor-binding domain, causing a change in the protein conformational state and rendering it incapable of binding to ST2 [183] (Figure 3D).

### 3.2. IL-33, a Multifunctional Cytokine during IAV Infection

IL-33 is constitutively expressed by AECs [142], although the source of IL-33 during viral infection seems to be host-dependent. Indeed, the source of IL-33 in humans seems to be ciliated AECs, with some positive IL-33 staining within airway basal cells. In contrast, in mice, IL-33 is mainly found in lung alveolar type II cells [35], as well as endothelial cells, along with infiltrating immune cells [34].

Evidence obtained in humans and mice suggests that IL-33 acts at multiple levels during IAV infection. Firstly, it regulates immune responses necessary for the anti-viral response and initiation of tissue repair post-IAV infection; secondly, it is necessary for the defense against bacterial infections post-IAV infection; and thirdly, it plays a central role in virus-induced asthma exacerbations [84]. Since mechanisms that control IAV infection and resolution of morbidity are similar in humans, mice, and horses, it is tempting to extrapolate the data obtained in these species to horses. However, the role of IL-33 in EIV pathogenesis still awaits formal scientific demonstration.

#### 3.2.1. IL-33 Regulates Immune Responses during IAV Infection

IL-33 has been shown to affect the host immune response positively and negatively during IAV infection. Discrepancies may depend on the lung pathophysiological status of the host prior to infection.

During IAV infection, increased levels of IL-33 protein and up-regulation of the *il-33* gene are observed in murine lungs and bronchoalveolar lavages. Overexpression of the *il-33* gene is positively correlated with a significant increase in IL-6, IFN-γ, IL-1β, and TNF-α mRNA, important inflammatory cytokines involved in the cytokine storm in severe IAV infections [84]. In mice models of IAV infection, exogenous IL-33 was shown to induce the recruitment of DC and increase the secretion of IL-12, promoting cytotoxic CD8+ T cell responses in the local microenvironment [184]. Indeed, although naïve CD8+ T cells express low levels of ST2, it is up-regulated after IL-12 stimulation, and IL-33 notably increases CD8+ T cells TCR-triggered IFN-γ production [164] and antiviral protective recall responses [165]. In vitro [185] and in vivo [186] studies have also shown that in synergy with IL-12, IL-33 regulates the expression of transcription factors linked to CD8+ TRM differentiation (induction of T-bet and Blimp-1 and repression Eomes and TCF-1) and participates in the formation and maintenance of lung CD8+ Resident Memory T Cells (CD8+ TRM) [165]. Furthermore, CD8+ TRM differentiation and maintenance are likely to depend on IL-33 levels in the lung tissue and the IL-33-ST2 signaling pathway [187]. Additionally, IL-33 has been described to enhance IL-12-induced NK and iNKT IFN-γ production [188,189].

The IL-33 cytokine signals through Myeloid differentiation primary response 88 (MyD88) and nuclear factor-kappa B (NF-κB) [31,147]. NF-κB is also important for production of type I IFNs during IAV infection. Interestingly, the IAV NS1 protein has been shown to block type I IFN production by antagonizing NF-κB activation [190,191,192]. Several studies have also shown that IAV NS1 antagonizing function of NF-κB varies between strain and subtypes [54,193]. Whether NS1 impacts NF-κB activation in the IL-33 pathway is not currently known.

IL-33 is also important during the recovery phase of IAV infection. Indeed, tissue repair post-IAV infection is mediated by IL-33-induced production of amphiregulin (AREG), a member of the epidermal growth factor (EGF) family, by immune cells such as Tregs and ILC2s [194,195,196]. AREG acts through the epidermal growth factor (EGF) receptor (EGFR) and promotes proliferation of airway epithelial progenitors, responsible for the repair of the damaged airway epithelium [184,194]. IL-33 also enhances the polarization of alternatively activated macrophages (AAM) [171,197], important in the resolution of inflammation and promotion of tissue repair post-IAV infection. Indeed, these macrophages are able to remove cellular debris and apoptotic cells after tissue injury and to avoid the further propagation of the inflammation by expressing suppressor receptors (such as PD-L1) and anti-inflammatory mediators (such as IL-10) [198,199].

Recent studies have also implicated IL-33 in virus-induced asthma exacerbations by driving type 2 immune responses, which is enhanced in human asthmatic patients.

#### 3.2.2. The Balance IL-33/Type I IFN Regulates IAV Disease Severity in Asthmatics

In humans, genome-wide association studies have associated asthma risk with variants in or near *il-33* and *st2* loci [200,201,202]. IL-33 expression is increased in the lungs and airways of patients with asthma and correlates with disease severity [139,140]. IL-33 levels are also increased in mice models of asthma, where its blockade is protective [203,204,205,206].

Acute worsening of asthma symptoms (also known as asthma exacerbations) are often caused by respiratory viruses, such as respiratory syncytial virus, rhinovirus, and IAV [132,134,135,136,137,207]. In the case of IAV-induced asthma exacerbations, several mechanisms have been proposed, including inflammation-induced increased levels of sialic acid expression in AECs (site of entry of IAVs) [208,209] and exaggerated type 2 immune responses [210,211]. In asthma mice models, exposure to the allergen house dust mite (HDM) prior to IAV infection also led to suppression of Th1-like innate and adaptive antiviral responses as well as cytotoxic responses in response to IL-33. Exposure to HDM also dampened AEC and dendritic cell (DC) type I and II IFN induction [35,212,213], thereby increasing viral loads [35,214].

Evidence obtained in human and mice have revealed that IL-33 is released early after viral challenge (within 2 days of infection), and predominantly in areas of acute lung inflammation where it drives immune cellular influx and de novo *il-33* gene expression in lung cells [35]. Rapidly after that, other mechanisms come into play in viral-induced asthma exacerbations, and IL-33 does not influence inflammation and recruitment of immune cell populations in the lung to the same magnitude [35]. In a mouse model of IAV-induced asthma exacerbation using the Sendai virus, it was also found that viral infection induced an early spike in *il-33* gene expression followed by a second more intense phase that persisted for several weeks after viral clearance [215]. These data implicated a switch from acute to chronic IL-33 release, which could be linked to the pro-inflammatory versus repair activity of IL-33 in IAV infection, and/or to be related to long-lived alterations to epithelial progenitor cells, as described in other respiratory chronic inflammatory conditions [215].

Furthermore, in asthmatics, IL-33 also suppresses type I IFN expression in both structural cells and DCs [35], although the exact mechanism awaits further studies. In asthma mice models, exposure to the allergen house dust mite (HDM) prior to IAV infection led to suppression of Th1-like innate and adaptive antiviral responses as well as cytotoxic responses in response to IL-33. Exposure to HDM also dampened AEC and dendritic cell (DCs) type I and II IFN induction, thereby increasing viral loads [35,214]. Whether this is also the case in other mammals, such as horses, awaits further research. Conversely, studies have found that expression of the type I IFN receptor (IFNAR) on human and mouse immune cells mediated suppression of type 2 cytokines that drive allergic immune responses in asthmatics [216]. Therefore, anti-viral and allergic responses seem to be counter-regulatory, adding complexity to the association between viral infections and asthma.

Aside from its role in viral infection, IL-33 is also important in the host defense against bacterial infection post-IAV infection.

#### 3.2.3. IL-33 Promotes Anti-Bacterial Host Defense during IAV Infection

The role of IL-33 in host defense against bacterial infections was first shown in murine sepsis and skin models [217,218,219]. In IAV-infected mice, attenuation of bacteria-induced IL-33 was shown to predispose to secondary bacterial pneumonia by reducing neutrophil function [36]. Earlier studies also showed a decrease in bacterial killing functions of neutrophils during IAV infection [112,220,221], which was driven by the suppression of nicotinamide adenine dinucleotide phosphate (NADPH) oxidase-dependent bacterial clearance [120]. However, whether or not IL-33 played a role in this phenomenon remains unknown.

The protective role of IL-33 against secondary bacterial infection post-IAV infection is further highlighted in asthmatics. Indeed, in humans, during the 2009 influenza pandemic [222] it was noted that asthmatics, who intrinsically produce higher levels of IL-33 in response to viral infection, had less chance to develop secondary bacterial pneumonia [223,224,225,226]. Furthermore, evidence obtained in murine airway allergy models showed that allergic mice infected with IAV had reduced *Streptococcus* bacterial burdens compared to non-allergic mice [138].

Whether secondary bacterial infections are less likely in asthmatic horses also remains an open question. This area of research is of particular importance for racehorses as the vast majority (80%) of Thoroughbreds in active training and racing have chronically inflamed airways [227].

Increasing our understanding of the impact of IL-33 in the control and resolution of IAV infection, using a One Health approach, is critical for the development of vaccines that promote prompt and healthy recovery in normal and asthmatic patients, being humans or horses.

### 3.3. IL-33 Is an Important Adjuvant for IAV Vaccines

While the role of IL-33 in IAV vaccine efficacy in humans or horses remains unknown, data obtained in murine models point towards an important role of this cytokine as a mucosal vaccine adjuvant. Indeed, a recent investigation of nasal alum-adjuvanted IAV vaccine in mice models induced a temporary release of IL-33 (within 24 h) by necroptotic alveolar epithelial cells, leading to enhanced antigen-specific Immunoglobulin A (IgA), but not IgE production. This effect was mediated by IL-33-driven production of IL-5 and IL-13 from ILC2s and enhanced MHC II expression in lung antigen-presenting cells (APCs), driving T cell activation [149]. Another study carried out in mice investigating the mechanisms of action of the Hydroxypropyl-β-Cyclodextrin adjuvant co-administered with IAV vaccines identified the IL-33/ST2 pathway as essential for adjuvant immunogenicity in intranasal, but not subcutaneous administrations [228]. The mechanisms described here are different from those reported for alum adjuvanticity in other routes of administration [229,230,231,232]. Besides this, a recent study comparing the combinatorial effects of exogenous IL-33 administration and endogenous IL-33 release by AECs during IAV infection in murine models showed enhanced protection to infection via the recruitment of DCs, increased secretion of IL-12, and enhanced cytotoxic CD8+ T cell responses [184]. Furthermore, after intranasal administration of recombinant IAV hemagglutinin (rHA) protein in mice [233], exogenous IL-33 was found to increase IgG and IgA responses in plasma and respiratory mucosa, respectively. This effect was notably driven by mast cells [233]. Intranasal administration of exogenous IL-33 was also shown to increase the IgA response of an inactivated influenza virus vaccine, in an ILC2- and potentially a Th2-dependent manner [234].

Investigation of the role of IL-33 in vaccine efficacy in natural IAV hosts, such as humans and horses, warrants further investigation for the development of safe IAV vaccines in susceptible populations, such as asthmatics, where production of protective antigen-specific IgA antibodies, without deleterious antigen-specific IgE antibodies, would be beneficial.

## 4. Conclusions and Perspectives

The outcome of IAV-associated lung injury in horses and humans is determined by both viral and host factors, suggesting an optimal range of activity for the immune response to viral infection.

Understanding how polymorphism in IAV non-structural NS1, PB1-F2, and PA-X proteins relate to AEC death type and disease severity, and how the lung pathophysiological status of a human or equine patient (healthy versus asthmatic) prior to IAV exposure affects the course of infection is critical to predict and prevent severe epithelial damage and pulmonary complications post-IAV infection. Using an integrated approach to health, the identification of disease-specific cytokines and pathways that can be specifically targeted or used as biomarkers to identify patient subsets in whom these targeted therapies will be most effective is necessary. Here we propose that the IL-33/ST2 pathway represents such a target. Indeed, modulating the IL-33 response during IAV infection would be a promising strategy for controlling lung inflammation and tissue damage, while avoiding secondary bacterial complications and promoting a healthy repair of the damaged respiratory epithelium. This strategy would also need to be tailored to the patient pathophysiological status to avoid any respiratory complications, such as asthma exacerbations. Given the acute IL-33 responses during IAV-induced exacerbation, local short-term treatment in the airways of patients after the very first signs of infection should be contemplated.

Experimental data obtained on currently available animal models of IAV infection (mainly mice and ferrets) have allowed investigators to elucidate critical pathways in IAV pathogenesis and helped predict the risk of severe epithelial damage and pulmonary complications in humans. However, the translation of knowledge obtained in these animal models to humans is somewhat limited by the fact that these species are not natural hosts of IAV, and they do not fully recapitulate the human disease. In addition, these species do not develop asthma naturally either. Improvements to these models, for example by using the horse, may be valuable in that aspect. Indeed, although infected by distinct IAV subtypes, horses are natural hosts of IAV. Humans and horses develop a similar clinical picture upon IAV infection (similar symptoms, course of infection, histological changes to the respiratory tissue, and immune responses to viral infection). Furthermore, several viral determinants of disease severity (notably in the IAV non-structural proteins) are shared between IAVs infecting horses or humans. Additionally, the complication of IAV infections in both humans and horses include secondary bacterial infections with *Streptococcus* and *Staphylococcus* species. Finally, similar to humans, horses naturally develop various forms of asthma, some of which share important common features with the human disease. However, increasing the knowledge about equine IL-33 and its role in immune responses and lung damage upon IAV infection will be essential before using the horse as a model for humans, and for a One Health approach to IAV.

## Figures and Tables

**Figure 1 viruses-13-02519-f001:**
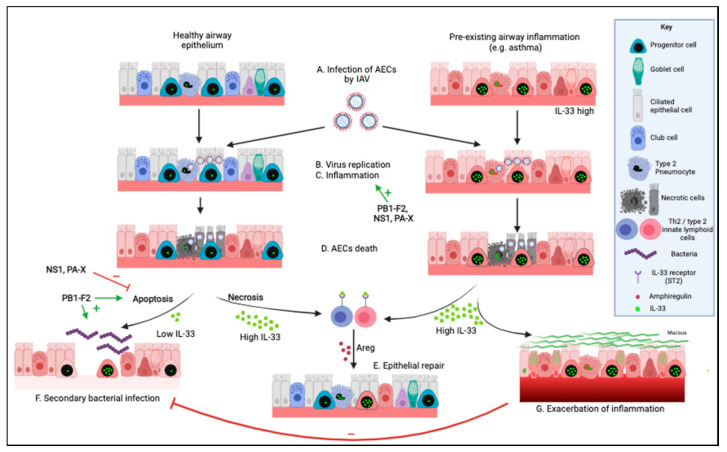
Effects of influenza A virus on the respiratory epithelium. In healthy individuals, (**A**) influenza A virus (IAV) infects airway epithelial cells (AECs). Viral replication (**B**) and/or inflammation (**C**) cause extensive AEC death (**D**). Evidence from other species indicates that the alarmin Interleukin-33 (IL-33) cytokine plays a key role in airway epithelial repair (**E**), by binding the IL-33 receptor on T regulatory cells (Tregs) and innate lymphoid cells type 2 (ILC2s) and inducing amphiregulin (AREG) production. In turn, AREG promotes airway progenitor proliferation and repair of the airway epithelium. Additionally, the death of AECs during an IAV infection can be of two types, apoptosis or necrosis, each directly impacting in opposing ways on the level of bioactive IL-33 released in the extracellular space. Apoptosis leads to inactivation of IL-33, thus limiting airway inflammation, but potentially increasing the risk of secondary bacterial infection (**F**). On the other hand, necrosis allows the release of high levels of bioactive IL-33, which is necessary for the prevention of secondary bacterial infection but promotes inflammatory responses (**G**) and exacerbation of chronic inflammatory lung disease, such as asthma. Interestingly, the non-structural proteins of IAV have been shown to impact virus replication and lung inflammation (NS1, PB1-F2, PA-X), as well as to regulate apoptosis in infected cells (NS1, PB1-F2, PA-X) and to promote secondary bacterial infection (PB1-F2). Here, we propose that IAV affects extracellular bioactive IL-33 levels and either promotes secondary bacterial infection (low IL-33) or increases tissue inflammation and exacerbation of chronic airway diseases (high IL-33) in a strain-dependent manner. Created with https://biorender.com/ (Accessed date 14 November 2021).

**Figure 2 viruses-13-02519-f002:**
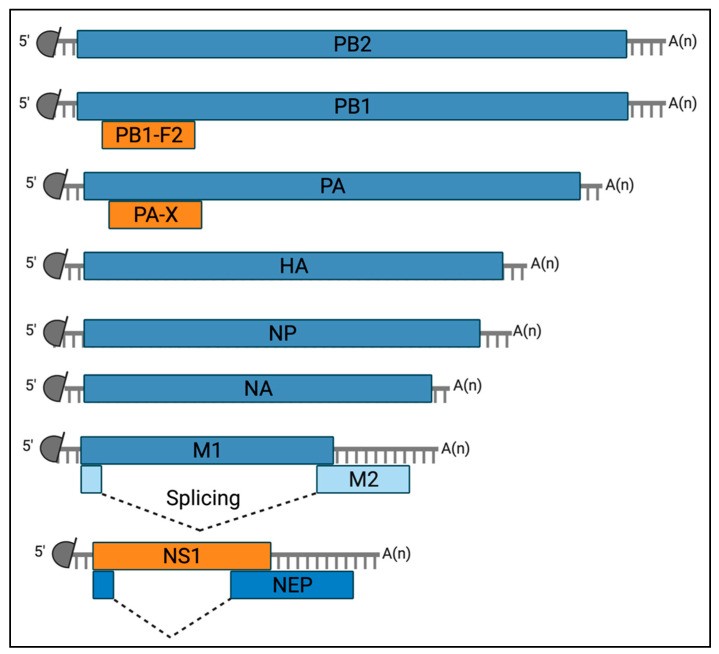
Diagram of the viral mRNAs transcribed from the EIV genome. Schematic representation of the twelve known mRNA molecules transcribed from the EIV genome. Grey semi-circles represent the 5′ methylated cap structure and grey lines denote host-derived primers of the 10 to 14 nucleotides that are obtained by the cap-snatching mechanism of the viral polymerase. A(n) corresponds to the 3′ poly-A tail produced by reiterative stuttering of the viral polymerase. Boxes indicate the viral gene product encoded by each mRNA. The constituent of the viral ribonucleocomplex (vRNP), including the three subunits of the viral polymerase (PB2, PB1, and PA) are encoded on segment 1 to 3, respectively, and the Nucleoprotein (NP) on segment 5. The membrane-associated glycoprotein of surface Hemagglutinin (HA) and Neuraminidase (NA) are encoded on segments 4 and 6, respectively. The matrix protein M1 and the ion channel matrix protein M2 are encoded on segment 7, on a colinear mRNA for the former and via alternative splicing (dashed lines) for the latter. The non-structural protein 1 (NS1) is encoded on a colinear mRNA derived from segment 8. This segment also encodes for the Nuclear Export Protein (NEP) mRNA via alternative splicing (dashed lines). Finally, two accessory proteins, PB1-F2 and PA-X, are encoded via partially overlapping open reading frames of segments 2 and 3, respectively. Created with https://biorender.com/ (Accessed date 14 November 2021).

**Figure 3 viruses-13-02519-f003:**
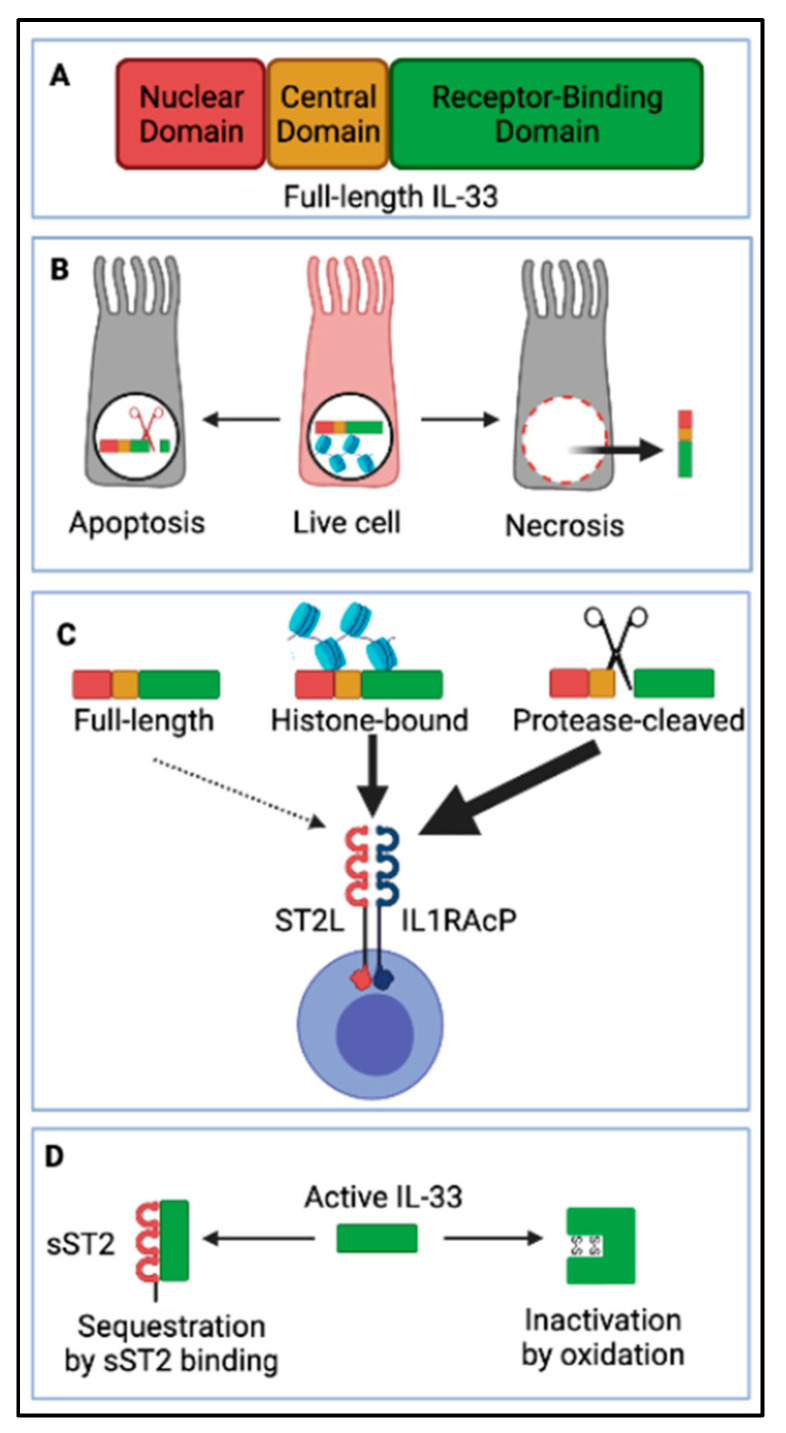
Interleukin-33 (IL-33) processing and release. (**A**) Full-length IL-33 consists of an N-terminal nuclear domain (ND), a protease-sensitive central domain (CD), and a C-terminal IL-1-like receptor-binding domain (RBD). (**B**) IL-33 is stored in the nuclear compartment of live airway epithelial cells, tethered to DNA. Upon cell necrosis, full-length IL-33 is released in the lung interstitium, while upon apoptosis it is cleaved in its RBD by caspases and thus inactivated. (**C**) Maturation of IL-33 is not essential for activity as full-length IL-33 is bioactive, however, proteases from mast cells, neutrophils, and allergens can cleave it in the protease-sensitive central domain, releasing shorter versions of IL-33 with even more potent activity. Additionally, full-length IL-33 can be released in complex with histones and show higher activity compared to full-length IL-33 alone. IL-33 signals through a receptor complex consisting of a transmembrane suppression of tumorigenicity 2 (ST2L) and IL1 receptor accessory protein (IL1RAcP) expressed by a variety of immune cells. (**D**) As IL-33 is a very potent alarmin cytokine, mechanisms to limit its activity are essential to avoid immunopathology. Aside from the previously mentioned (**B**) tethering of full-length IL-33 to DNA preventing its secretion in live cells, and inactivation by intracellular caspases during apoptosis, IL-33 can be sequestrated by a soluble version (decoy) of its receptor sST2 or inactivated by oxidation due to formation of disulfide bonds, which modify the protein conformation. Created with https://biorender.com/ (Accessed date 14 November 2021).

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
