# Peer review of "Could Interleukin-33 (IL-33) Govern the Outcome of an Equine Influenza Virus Infection? Learning from Other Species"

_viruses, 2021, doi:10.3390/v13122519_

Round 1
Reviewer 1 Report
In the review by Rozario et. al. Equine influenza virus infections are discussed with a focus on viral proteins with immune evasion mutations that impact viral pathogenesis. The review focuses in on the role of alarmin or IL-33 in influenza pathogenesis and argues that more knowledge is needed in the equine model system.
Major concerns regarding the manuscript in its current form are as follows:
- The authors state on line 390 that horses express an IL-33 protein that shares homology with mouse and human IL-33, but little else is known. Given that the authors argue that observations regarding IL-33's role in equine influenza are needed and that the majority if not all of the literature discussed in the review regarding IL-33 is from studies reporting findings in humans or that use murine models, the title of the current review is not appropriate and somewhat misleading. A more fitting title is required.
- To follow up on the previous point, changes are needed in the abstract so that it more truly summarizes the review content.
- Figure 1 is a cartoon of alveolar epithelial cell infection by equine influenza virus, but is based on human and murine findings per point #1. Reference to IAV in generic terms in the figure as well as figure legend is more appropriate.
- It would be beneficial if the discussion of the need for more study beyond the knowledge of IL-33 expression in horses is moved to the paragraph prior to section 3 and starting on line 309.
Reviewer 2 Report
The health of animals, humans, plants, and the state of ecosystems is interconnected. Only an integrated approach that takes into account this fundamental relationship allows us to counter threats to the health of animals, people, plants, and the environment. This review article will help us to move towards an integrated approach to health. The article is well written and excellently illustrated. I am more and more convinced of how good the BioRender web service is and what great illustrations it can create. From my point of view, only a few minor corrections should be made.
Point 1: Line 533-537. I deeply respect the One Health program that attains optimal health for people, animals, and our surroundings; however, this sentence is not clear. How understanding the lung pathophysiological status of healthy and asthmatic humans with human flu may help to predict and prevent severe epithelial damage and pulmonary complications in horses with equine flu? Please speak more clearly about people and horses and specify which discoveries in the field of animal virology can be extrapolated to humans and vice versa. It would be more logical to state that experimental data obtained on animals can help to predict and prevent severe epithelial damage and pulmonary complications in humans and not vice versa. Please, explain.
Point 2: One Health approach is mentioned in the text-only once in passing (not counting one more mention in the abstract). I believe that it should be given a more detailed description of the One Health approach, or deleted altogether. Then there will be no question why the authors "jump" from the description of equine flu to human flu and vice versa.
Point 3: When talking about the role of IL-33 in the inflammatory process in humans and horses (for instance, see the sentence in Lines 502-504), probably, the description of the impact of IL-33 in the control and resolution of flu should have been preceded by the statement that the mechanisms of its action in both humans and horses are more or less the same which allows extrapolating the data obtained.
Point 4: Section 3.3 “IL-33 is an important adjuvant for IAV vaccines.” Please, add the model(s). I guess it should be a mouse model.
Point 5: Section 3.3 “IL-33 is an important adjuvant for IAV vaccines.” In almost all of the cited publications, studies were carried out on mice. Could you please explain what is the connection between this section and the main goal and the title of the article, which mentions equine flu? Perhaps this will become clearer by moving the paragraph (lines 525-529) to the beginning of this section.
Point 6: This comment is in some way related to Point 5. Please read your review paper carefully in terms of how many sections are dedicated to equine influenza and how many experimental data from other animals or humans were used. Please, consider changing the title or removing equine influenza from there.
